# Oxidative Stress in Liver Pathophysiology and Disease

**DOI:** 10.3390/antiox12091653

**Published:** 2023-08-22

**Authors:** Abdolamir Allameh, Reyhaneh Niayesh-Mehr, Azadeh Aliarab, Giada Sebastiani, Kostas Pantopoulos

**Affiliations:** 1Department of Clinical Biochemistry, Faculty of Medical Sciences, Tarbiat Modares University, Tehran 1411713116, Iran; allameha@modares.ac.ir (A.A.); r.niayeshmehr@modares.ac.ir (R.N.-M.); azadehaliarab@modares.ac.ir (A.A.); 2Chronic Viral Illness Services, McGill University Health Center, Montreal, QC H4A 3J1, Canada; giada.sebastiani@mcgill.ca; 3Department of Medicine, McGill University, Montreal, QC H4A 3J1, Canada; 4Lady Davis Institute for Medical Research, Montreal, QC H3T 1E2, Canada

**Keywords:** ROS, liver disease, fibrosis, hepatitis, NAFLD, NASH, hepatocellular carcinoma

## Abstract

The liver is an organ that is particularly exposed to reactive oxygen species (ROS), which not only arise during metabolic functions but also during the biotransformation of xenobiotics. The disruption of redox balance causes oxidative stress, which affects liver function, modulates inflammatory pathways and contributes to disease. Thus, oxidative stress is implicated in acute liver injury and in the pathogenesis of prevalent infectious or metabolic chronic liver diseases such as viral hepatitis B or C, alcoholic fatty liver disease, non-alcoholic fatty liver disease (NAFLD) and non-alcoholic steatohepatitis (NASH). Moreover, oxidative stress plays a crucial role in liver disease progression to liver fibrosis, cirrhosis and hepatocellular carcinoma (HCC). Herein, we provide an overview on the effects of oxidative stress on liver pathophysiology and the mechanisms by which oxidative stress promotes liver disease.

## 1. Major Types of Liver Cells and Their Role in Liver Functions

The liver is a large organ constituting about 2% of body weight in adult humans [1]. It is anatomically divided into a larger right and a smaller left lobe, each made up of thousands of lobules. The liver lobules contain parenchymal and non-parenchymal cells that interact to form a functional hepatic unit (Figure 1) [2]. Parenchymal cells include hepatocytes, the predominant liver cell population, and cholangiocytes. Non-parenchymal cells include sinusoidal endothelial cells, macrophages, stellate cells and natural killer cells [3,4].

### 1.1. Hepatocytes

The liver parenchyma is mostly composed of hepatocytes, which make up to 80% of total liver cells [5]. They perform vital functions such as the clearance of toxic metabolites and xenobiotics, as well as the secretion of proteins and lipids to maintain blood homeostasis. Hepatocytes also produce hormones [6] and bile [7], and mediate innate immune responses [8]. Nutrient-rich blood from the portal vein and oxygenated blood from the hepatic artery are directed to hepatocytes via highly specialized capillaries known as hepatic sinusoids [9]. In liver lobules, hepatocytes are arranged into plates around the central vein, which are separated by liver sinusoids. To enhance absorption from the plasma, hepatocytes extend a number of microvilli into the space of Disse, a thin peri-sinusoidal region between the hepatocytes and sinusoidal endothelium [10].

### 1.2. Cholangiocytes

Cholangiocytes are specialized epithelial cells lining the extra and intrahepatic bile ducts, which make up 3–5% of liver cells [11]. They are involved in the synthesis (together with hepatocytes), secretion and modification of bile, which is essential for the digestion and absorption of fats. Cholangiocytes also regulate bile flow in response to hormonal and neural signals and exhibit immune functions. They represent a heterogeneous cell population of biochemically and morphologically distinct large and small cholangiocytes.

### 1.3. Liver Sinusoidal Endothelial Cells (LSECs)

LSECs are the most abundant type of liver non-parenchymal cells, constituting about 15–20% of the total liver cell content [12]. They are very specialized endothelial cells that discontinuously line the wall of hepatic sinusoids and make a unique permeable barrier between blood within the sinusoids and the underlying hepatocytes and hepatic stellate cells. LSECs have distinctive morphological features, including the presence of numerous small open pores called fenestrae, and the absence of a classical basement membrane and diaphragm, which help their permeability. LSEC fenestration provides access of hepatocytes to certain substances from the circulating blood by connecting the lumen of the sinusoids with the space of Disse. In addition, LSECs have important physiological and immunological functions such as the regulation of hepatic vascular tone by synthesizing nitric oxide (NO) and endothelin-1 (ET-1), the maintenance of hepatic stellate cells and Kupffer cells in a quiescence state, filtration, endocytosis, antigen presentation and leukocyte recruitment. LSECs are the first liver cells to be affected by liver injury and contribute to the initiation and progression of liver disease. Thus, following liver injury, LSECs undergo morphological changes culminating in capillarization (defenestration), which causes sinusoidal endothelial dysfunction [13,14].

### 1.4. Hepatic Stellate Cells (HSCs)

HSCs, also known as Ito cells, are non-parenchymal liver pericytes located in the space of Disse [15,16]. They store 50–95% of the body’s vitamin A and can communicate with all cell types within the liver, either physically or through cytokines and chemokines. HSCs are considered liver-specific mesenchymal stem cells (MSCs), as they possess properties akin to stem/progenitor cells. In the healthy liver, quiescent HSCs play a vital role in vitamin A homeostasis. They also contribute to the regulation of extracellular matrix (ECM) turnover, immunoregulation, expression of growth factors required for liver development and regulation of hepatic blood flow with their contractile ability. Moreover, HSCs are critical for liver regeneration and repair [17]. In response to liver injury, quiescent vitamin A-rich HSCs undergo activation or trans-differentiation into proliferative and contractile myofibroblast-like cells, which is a key event in liver fibrosis [18]. Activated HSCs lose their vitamin A content and express α-smooth muscle actin (α-SMA; a key HSC activation marker), while they synthesize and release specific ECM components, such as collagens, proteoglycans and glycoproteins [18].

### 1.5. Liver Macrophages

Liver resident macrophages, also known as Kupffer cells (KCs), are located in the lumen of sinusoids adherent to LSECs [19]. They constitute the largest population of mononuclear phagocytes in the body and are considered a filter for gut-derived pathogens within the portal circulation. In physiological states, KCs contribute to immune surveillance by eliminating circulating pathogens and hazardous materials via phagocytosis. In addition, KCs play a key role in systemic iron homeostasis by clearing senescent red blood cells via erythrophagocytosis, which is coupled with the recycling of iron to the bloodstream for de novo erythropoiesis [20]. KCs are also crucial for cholesterol and lipid metabolism. Liver macrophages were earlier considered to consist of a single population of KCs. However, the use of advanced single-cell and spatial transcriptomics technologies uncovered an unexpected heterogeneity of liver macrophage populations [21]. Under pathological conditions, KCs and other liver residents, as well as recruited macrophages, are activated through various inducers and can acquire distinct phenotypes that affect liver disease outcomes [22].

### 1.6. Liver Natural Killer (NK) Cells

Liver resident NK cells, also known as Pit cells, are primarily found in the sinusoidal spaces. They are strategically positioned to monitor and respond to various substances, including pathogens and tumor cells, that enter the liver through the bloodstream. Pit cells exhibit morphological features of large granular lymphocytes and differ from circulating NK cells in several aspects. For instance, they are physically connected with their pseudopodia to the microvilli of hepatocytes in the space of Disse. Pit cells are a major part of the hepatic innate immune system, participate in host defense mechanisms against tumors and microbes, and are critical for maintaining immune balance [23].

## 2. Oxidative Stress in the Liver

Reactive oxygen species (ROS) such as superoxide radical (O_2_•^−^) and hydrogen peroxide (H_2_O_2_) emerge from the incomplete reduction of molecular oxygen. They are physiologically produced in mitochondria during aerobic respiration and are also generated in cells via enzymatic reactions. H_2_O_2_ can oxidize ferrous (Fe^2+^) to ferric (Fe^3+^) iron according to Fenton chemistry, which yields the short-lived but highly toxic hydroxyl radical (OH•) [24]. Ferric iron undergoes redox cycling and is subsequently reduced to ferrous by O_2_•^−^. Thus, in the presence of O_2_•^−^ and H_2_O_2_, iron acts as a catalyst for the generation of OH•, which may give rise to other reactive free radicals and propagate lipid peroxidation. On the other hand, O_2_•^−^ can react with nitrogen monoxide radical (NO•) to yield peroxynitrite (NOO^−^), a potent oxidant. Reactive free radicals and non-radical oxidants can attack and damage all cellular macromolecules, including proteins, nucleic acids and lipids. Protein oxidation may have negative functional consequences, while oxidation of nucleic acids may lead to mutagenesis. Peroxidation of membrane lipids, especially polyunsaturated fatty acids (PUFAs), may promote ferroptosis, an iron-dependent form of cell death [25].

ROS such as O_2_•^−^ and OH• are unstable free radicals. They owe their reactivity to their capacity to extract an electron from, or donate their unpaired electron to neighboring molecules, and thereby acquire a thermodynamically stable state. By contrast, H_2_O_2_ or NOO^−^ are stable non-radical oxidants. While ROS were initially considered as biohazards, it is now clear that at low levels they also act as second messengers and play a major role in physiological signaling pathways, gene expression regulation, host defense against microorganisms, immune responses and vasodilation. However, an uncontrolled rise of ROS levels is toxic and promotes “oxidative stress”, which has been defined as a disruption of the oxidant/antioxidant balance in favor of the former that can lead to tissue damage [26]. This pathologic state is lately also referred to as “oxidative distress”, to distinguish it from homeostatic “oxidative eustress” [27]. The same concept applies to NO• and other reactive nitrogen species (RNS): at low levels, they act as important second messengers, while in excess they disrupt the redox balance causing “nitrosative stress” [28].

The liver is an important site of ROS production by virtue of its metabolic and detoxification activities. ROS are generated via the mitochondrial respiratory chain and from other sources including peroxisomes, xanthine oxidases, cytochrome P450 oxidases and NADPH oxidases (NOXs) (Figure 2) [29]. Oxidases of the cytochrome P450 (CYP) family, such as ethanol-induced cytochrome P450 2E1 (CYP2E1), are involved in xenobiotics metabolism and constitute major sources of ROS in hepatocytes. Oxidases of the NOX family utilize NADPH and molecular oxygen to produce O_2_•^−^, which is rapidly dismutated to H_2_O_2_. The NOX2 isoform was originally characterized in phagocytes but is also expressed in KCs. Hepatocytes, LSECs and HSCs express various isoforms including NOX2 and the non-phagocytic NOX1, NOX4 and NOX5, as well as the dual oxidases DUOX1 and DUOX2 [30,31]. NOX4 has a unique mode of action compared to other NADPH oxidases and is known to predominantly generate H_2_O_2_ instead of O_2_•^−^ [32].

## 3. Antioxidant Defense Mechanisms in the Liver

Liver cells possess robust antioxidant defense mechanisms consisting of enzymatic and non-enzymatic components, which enable them to keep ROS at physiological levels. Antioxidant enzymes include superoxide dismutases (SOD), cytosolic glutathione peroxidases (GPX), glutathione reductases (GRX), peroxiredoxins (PRX), thioredoxins (TRX) and catalase (CAT). Glutathione (GSH), bilirubin, ubiquinone (coenzyme Q10), uric acid and vitamins E, A and C are established non-enzymatic antioxidants [33]. Dietary antioxidants, such as curcumin, resveratrol, quercetin and other flavonoids are also thought to contribute to protection against oxidative stress [34].

In general, enzymatic antioxidants metabolize ROS, while non-enzymatic antioxidants prevent or attenuate oxidative damages by neutralizing free radicals and non-radical oxidants. Inhibition of the activity or expression of enzymes involved in free radical production, such as NOXs and xanthine oxidases, or increasing the activity or expression of intracellular antioxidant enzymes is also part of the antioxidant defense armamentarium [35].

The NRF2/ARE signaling pathway (Figure 2) is considered the main cellular antioxidant defense mechanism [36] and has critical functions in liver pathophysiology [37]. NRF2 (nuclear factor erythroid 2-related factor 2) is a member of the Cap’n’Collar (CNC) basic leucine zipper (bZIP) family of transcription factors. Its primary function is to regulate the expression of a wide array of antioxidant and detoxification genes. Under normal physiological conditions, NRF2 is sequestered in the cytoplasm by its inhibitor, Kelch-like ECH-associated protein 1 (Keap1). Keap1 acts as a sensor for oxidative stress and facilitates the degradation of NRF2 via the proteasomal pathway. However, when cells encounter oxidants or electrophilic insults, specific cysteine residues on Keap1 are modified, leading to conformational changes in the Keap1–NRF2 complex. This results in the liberation and nuclear translocation of NRF2. Once in the nucleus, NRF2 forms a heterodimer with small musculoaponeurotic fibrosarcoma (MAF) proteins and binds to antioxidant response elements (AREs) present in the promoter regions of target genes [36].

The binding of NRF2 to AREs initiates the transcription of a battery of cytoprotective genes, including antioxidant enzymes (e.g., SOD, CAT and GPX), phase II detoxifying enzymes (e.g., glutathione S-transferases, NAD(P)H quinone oxidoreductase 1) and other stress response proteins. These gene products contribute to the cellular defense against oxidative stress by enhancing the intracellular antioxidant capacity and by promoting detoxification processes. The NRF2/ARE pathway is operational in parenchymal, as well as non-parenchymal, cells of the liver [37].

## 4. The Impact of Oxidative Stress on Liver Cells

The liver is continuously exposed to different toxic and reactive metabolites including ROS and RNS. A shift in the redox balance toward oxidative stress can be considered an initial step in the pathogenesis of liver diseases [38]. This process is affected by comorbidities such as diabetes/insulin resistance and by various exogenous factors such as alcohol abuse, viral infection, drug overdose, high-caloric diet, and exposure to environmental toxins, UV light or heavy metals. A surge in ROS and RNS levels is important in the onset of inflammatory reactions, fibrosis, necrosis, apoptosis or malignant transformation [28,38].

Hepatocytes are important sites of ROS production, especially in mitochondria, and are also sensitive to ROS-mediated injury. Each hepatocyte contains 1000 to 2000 mitochondria occupying about 20% of the cell volume [39]. ROS-mediated damage of lipids and particularly, PUFAs, can alter cell membrane fluidity and permeability. Mitochondrial lipid peroxidation negatively affects the electron transport chain, aggravating ROS production and oxidative stress [38]. Mitochondrial dysfunction in hepatocytes has been linked to the development and progression of chronic liver disorders [38,40]. For instance, patients with non-alcoholic steatohepatitis (NASH) exhibit hepatic oxidative stress due to impaired mitochondrial respiratory capacity and proton leakage [41]. In a mouse model of fatty liver disease, pharmacological improvement of mitochondrial redox homeostasis with the flavonoid dihydromyricetin was shown to be hepatoprotective [42]. Hjv^−/−^ mice, a model of hereditary iron overload (hemochromatosis), exhibit mitochondrial hyperactivity in hepatocytes, which predisposes them to HCC [43]. On the other hand, pharmacological chelation of mitochondrial iron has been shown to promote mitophagy, which protects mice against HCC [44]. Experiments with primary rat hepatocytes and rat H4IIEC3 hepatoma cells showed that palmitate treatment promotes a flux of calcium from ER to mitochondria, which causes mitochondrial oxidative stress and lipotoxicity. These data highlight the physiological importance of finetuning mitochondrial activities and redox balance. Accumulation of ROS can induce hepatocellular dysfunction or death that will eventually result in the release of damage-associated molecular patterns (DAMPs). Under these circumstances, non-parenchymal cells, such as KCs, HSCs and newly recruited immune cells are activated and produce pro-fibrogenic and pro-inflammatory mediators [45].

Oxidative stress promotes an influx of calcium into cells and redistribution of cellular calcium from the endoplasmic reticulum (ER) to the cytosol, mitochondria and nuclei, which in turn may trigger apoptotic and necrotic death [46]. These responses increase mitochondrial permeability transition and facilitate the release of pro-apoptotic factors such as cytochrome c, and the activation of calcium-dependent endonucleases, proteases and lipases, contributing to the death of hepatocytes and other liver cell types [47]. In addition, oxidative stress can affect the secretory functions of hepatocytes by disrupting the formation of bile flow, leading to cholestasis [48].

While ROS and lipid peroxidation products impair hepatocellular function and viability, they promote the differentiation and activation of HSCs to myofibroblasts, leading to the secretion and accumulation of collagen and other ECM components within the liver [49]. Therefore, chronic activation of HSCs in response to oxidative stress favors the development of liver fibrosis, which may progress to cirrhosis and HCC [50,51].

KCs are also activated by certain stimulants leading to ROS production, expression of a variety of cytokines and pro-inflammatory mediators, and recruitment of more immune cells [52]. Experimental studies with animal models have shown that ROS originating from KCs play a prominent role in the development of liver injury in response to hepatotoxins [53,54,55]. KCs secrete transforming growth factor β (TGF-β) and platelet-derived growth factor (PDGF), which in turn promote HSC activation, contributing to liver fibrosis [52]. In addition, KCs can directly kill hepatocytes through the activation of Fas-dependent apoptosis [56].

LSECs are sensitive to oxidative stress mainly due to their low H_2_O_2_ clearance capacity [57,58], but also due to their exposure to gut-derived toxins carried in the portal vein [59]. Therefore, ROS can selectively damage LSECs and impair their physiological activities. For instance, the oxidation of spectrin can disrupt its interaction with actin, which is essential to maintain fenestrae structure and function, and thereby cause fenestrae closure [60]. Defenestration impairs the bidirectional exchange of molecules between hepatocytes and hepatic blood sinuses [61]. Vascular endothelial dysfunction driven by oxidative stress and inflammation plays an important role in liver injury [62,63,64]. Thus, it may lead to the decreased generation of vasodilator factors such as NO and promote vasoconstriction, which causes increased resistance in sinusoidal microcirculation and portal hypertension [65].

Autophagy is the main endogenous recycling process that preserves cell homeostasis under physiological conditions and offers a survival mechanism under stress [66]. Experimental studies with animal models suggest that autophagy protects LSECs against oxidative stress responses to acute liver injury, while impairment of this pathway leads to endothelial dysfunction and contributes to HSC activation and liver fibrosis [62]. Along these lines, NASH patients exhibit smaller autophagic vacuoles in LSECs [67], indicating that autophagy is dysregulated in liver diseases.

Cholangiocytes are involved in cholangiopathies, which can be mediated by oxidative stress factors [48]. Nevertheless, the effects of oxidative stress on cholangiocyte pathophysiology are not well understood. As an example, increased oxidative stress can induce senescence in cholangiocytes through the stimulation of ER stress [68]. On the other hand, melatonin appears to protect cholangiocytes against oxidative stress-induced cell damage and inflammation [69].

NK cells are generally susceptible to ROS, and oxidative stress can alter their activity. This can contribute to immune escape within the tumor microenvironment [70,71]. However, the role of oxidative stress on liver resident NK cells (Pit cells) in the context of liver disease is not clear. The effects of oxidative stress on the various liver cell types are schematically illustrated in Figure 3.

## 5. Role of NOX Enzymes in Oxidative Damage to Liver Cells

NOX enzymes are major sources of ROS production in the liver [72]. Moreover, it appears that NOX-derived ROS are important contributors to the onset and progression of liver injury and chronic liver disease [30]. Thus, the activation of NOX1, NOX2 and NOX4 isoforms in liver cells is associated with HSCs activation, apoptosis/necrosis of hepatocytes and the amplification of inflammatory responses through KCs activation [73,74,75].

NOX-dependent ROS production is tightly linked to the TGF-β signaling pathway [76], which drives various liver pathologies, especially fibrosis and cancer [77]. TGF-β causes redox imbalance by directly increasing the production of ROS or by downregulating the expression of antioxidant enzymes. Conversely, ROS can induce TGF-β activation and stimulate TGF-β-related functions [78,79]. The degree of TGF-β activation reflects the severity of liver injury and fibrosis. Hepatocellular NOX4 is a downstream target of TGF-β1 and plays a crucial role in oxidative stress-induced apoptosis [80]. Experiments in mouse models showed that during liver injury, NOX4 [81,82,83], as well as NOX1 [84] and NOX2 [85], is activated in HSCs and promotes fibrogenesis. Under these conditions, the activation of NOX enzymes in HSCs is mediated by profibrogenic agonists such as angiotensin II (Ang II) and PDGF, or by the phagocytosis of apoptotic bodies or other cellular debris from dead hepatocytes.

KCs express NOX2, which generates ROS and thereby enhances production of pro-inflammatory cytokines (such as TNFα, IL-6 and IL-1β). The pro-inflammatory cytokines induce infiltration of neutrophils and, thus, indirectly trigger the activation of HSCs [86].

## 6. Oxidative Stress in Drug-Induced Liver Injury

Drug-induced liver injury can lead to a rapid decline in liver function and acute liver failure [87,88]. Hepatotoxic drugs may cause the accumulation of ROS/RNS and induce oxidative/nitrosative stress in the liver by different mechanisms, including the increase in intracellular oxidants, lipid peroxidation, depletion of antioxidants and mitochondrial dysfunction. Therefore, the dysregulation of redox balance is a hallmark of drug-induced liver injury. A typical example is provided by acetaminophen (N-acetyl-p-aminophenol) toxicity. This drug (widely known as paracetamol) is metabolized by cytochrome P450 enzymes (mainly CYP2E1 and CYP1A2) to a reactive intermediate, N-acetyl-p-benzoquinone imine, which is detoxified following conjugation with GSH. However, excessive acetaminophen intake eventually leads to the depletion of the GSH pool of the liver, causing the death of hepatocytes and innate immune activation [89]. Pharmacological administration of N-acetylcysteine (NAC), a GSH precursor, can prevent and reverse acetaminophen liver injury. In fact, NAC is clinically used as an antidote.

## 7. Oxidative Stress in Chronic Viral Hepatitis

Hepatitis is an inflammatory condition of the liver caused by various exogenous and endogenous factors, including chronic alcohol intake, drugs, toxins, autoimmune disorders or viral infection. The severity of hepatitis can vary from mild and self-resolving to severe. Chronic hepatitis may lead to complications including liver fibrosis, cirrhosis, cancer and/or liver failure [90]. Viral hepatitis is primarily caused by hepatotropic hepatitis A, B, C, D or E viruses (abbreviated as HAV, HBV, HCV, HDV and HEV, respectively) [91]. In addition, several other viruses are capable of inducing liver inflammation, such as Epstein–Barr virus, Herpes simplex virus and Cytomegalovirus [92]. Nevertheless, the major cause of chronic viral hepatitis is infection by HBV or HCV, and, in fewer cases, by HDV or HEV.

Chronic hepatitis B or C predisposes to the development of liver fibrosis and HCC [93,94,95]. It is estimated that more than 50% of HCC cases worldwide are associated with HBV and 25% with HCV infection [94]. There is evidence that these viruses contribute to liver disease by inducing oxidative stress and activating ROS-sensitive signaling pathways and inflammatory responses [93,96]. Thus, the HBx protein of HBV induces oxidative stress due to mitochondrial dysfunction [97]; it is highly expressed in HCC tissues and promotes hepatocarcinogenesis, even without liver fibrosis [95,98,99]. The HCV core protein is known to inhibit the mitochondrial electron transport chain and decrease intracellular and mitochondrial GSH levels [100,101], thereby causing oxidative stress. In line with these data, transgenic expression of this protein in mice causes HCC via oxidative stress and in the absence of inflammation [102,103]. Transgenic expression of the complete HCV transgene likewise causes HCC in mice by oxidative stress due to iron overload [104], again, independent of inflammation [105].

## 8. Oxidative Stress in Fatty Liver Disease

Increased deposition of lipids (steatosis) in the liver is the hallmark of non-alcoholic fatty liver disease (NAFLD) and alcoholic fatty liver disease (AFLD) [106]. Abnormal accumulation of fat droplets, mainly in the form of triglycerides, occurs in 5% or more of the hepatocytes and exhibits a microvesicular or macrovesicular histological pattern. Hepatic steatosis originates from abnormalities in lipid metabolism pathways including enhanced free fatty acid (FFA) uptake and de novo lipogenesis, combined with decreased triglyceride hydrolysis, fatty acid beta oxidation and lipid clearance [107]. Fatty liver disease was originally described in 1980 as a single entity [108]. However, NAFLD and AFLD were later considered as diseases with common pathophysiological and molecular features but distinct etiologies; they constitute the most common causes of chronic liver disorders worldwide [106,109,110]. In a subset of patients, liver steatosis progresses to steatohepatitis, which may further progress to fibrosis, cirrhosis and HCC. In steatohepatitis, including non-alcoholic steatohepatitis (NASH), excessive fat deposition in the liver is associated with necroinflammation and morphological alterations (ballooning) of hepatocytes. NAFLD is tightly linked to metabolic abnormalities, such as abdominal obesity, type 2 diabetes, dyslipidemia and hypertension. In fact, NAFLD is considered the hepatic manifestation of metabolic syndrome. To emphasize this, a group of international experts proposed in 2020 to rename NAFLD as “metabolic dysfunction-associated fatty liver disease” (MAFLD), without exclusion of alcohol consumption or viral hepatitis [111,112]. The term MAFLD is not based on a negative definition (non-alcoholic) and takes into account that the course and progression of the disease may be affected by alcohol consumption or viral infection. More recently, leaders from multinational liver societies proposed the non-discriminatory nomenclature “metabolic dysfunction-associated steatotic liver disease” (MASLD) and the use of “metabolic steatohepatitis” (MASH) instead of NASH [113,114,115]. Nevertheless, the terms NAFLD and NASH are still in use.

At the molecular level, fat accumulation in the liver is linked to ER stress, alterations of lipid metabolism and disruption of the autophagy pathway, leading to lipotoxicity [107]. Recent data have suggested that nicotinamide N-methyltransferase (NNMT) is an important contributor to liver steatosis following chronic alcohol consumption. This enzyme is expressed in hepatocytes and catalyzes S-adenosylmethionine (SAM)-dependent methylation of nicotinamide to 1-methylnicotinamide, which prevents nicotinamide adenine dinucleotide (NAD^+^) regeneration [116]. Thus, NNMT is a crucial regulator of NAD^+^ homeostasis. Chronic exposure of mice to alcohol was shown to increase hepatic NNMT expression via the ER stress-induced PERK-ATF4 pathway [117]. This response increased lipogenesis and promoted liver steatosis but alleviated liver injury [117,118]. NNMT upregulation has been shown to inhibit oxidative stress-induced autophagy [119,120]. On the other hand, endothelial NNMT appears to protect against oxidative stress and enhance endothelial cell viability [121].

It should be noted that progression of simple liver steatosis to steatohepatitis requires the action of multiple insults, one of them being oxidative stress [122,123]. In fact, mitochondrial dysfunction, lipid peroxidation and oxidative DNA damage have been demonstrated in the liver from NASH patients [124,125,126,127]. Interestingly, there is evidence from clinical trials and pilot studies that vitamin E, a lipophilic antioxidant, reduces levels of serum transaminases (ALT and AST) and improves liver histology in NAFLD/NASH patients [128,129,130]. A meta-analysis confirmed that vitamin E treatment significantly reduces ALT, AST and body mass index (BMI) in NAFLD patients, but it did not decrease the fibrosis score and total cholesterol [131]. Similar results were obtained in a meta-analysis of studies with pediatric NAFLD patients [132].

The role of oxidative stress in NASH pathogenesis is also highlighted in data obtained with animal models [133,134,135]. Interestingly, the amelioration of mitochondrial dysfunction was shown to delay NASH progression by altering the intestinal microbiome in mice [136]. While oxidative stress promotes ATP depletion in hepatocytes and may lead to cell death, the development of NASH is also linked to the activation of non-parenchymal liver cells, such as macrophages and HSCs. KCs and other non-resident liver macrophages undergo reprogramming to a proinflammatory phenotype with NASH-associated molecular signatures in humans and mice [137,138].

NAFLD is often associated with moderate hepatic iron overload, a known inducer of oxidative stress, which may affect progression to NASH [139]. Nevertheless, Hjv^−/−^ mice with severe iron overload in hepatocytes develop liver steatosis in response to a high fat diet but this does not progress to NASH [140,141]; notably, these animals exhibit lipid peroxidation and mitochondrial hyperactivity in the liver [43,142]. On the other hand, genetically obese Lepr^ob/ob^ mice accumulate excess iron in KCs in response to a high iron diet and develop histological signs of NASH [143]. In line with this, iron loading of KCs promotes their polarization to a proinflammatory M1 phenotype that drives the progression of NAFLD to NASH [144]. A recent study showed that in NAFLD/NASH livers, hepatocytes release iron to neighboring HSCs via extracellular vesicles, and the redistribution of iron contributes to lipogenesis, insulin resistance and fibrosis [145]. These findings highlight the different effects of iron-induced oxidative stress in parenchymal vs. non-parenchymal cells of the liver in the progression of NAFLD to NASH, at least in mouse models. They may also be relevant to humans, as NAFLD patients exhibit different histological patterns of iron overload, which are linked to disease pathophysiology [146].

## 9. Oxidative Stress in Genetic Liver Disorders

The most important genetic disorders leading to liver diseases include hereditary hemochromatosis, Wilson’s disease and alpha-1 antitrypsin deficiency; they are all transmitted in an autosomal recessive pattern.

Hereditary hemochromatosis is a genetically heterogenous endocrine disorder characterized by excessive dietary iron absorption, resulting in tissue iron overload [147]. Excess iron primarily accumulates in hepatocytes and to some extent also in parenchymal cells of the pancreas and heart. The cause of hereditary hemochromatosis is mutations in genes that regulate expression of hepcidin, the iron regulatory hormone. The predominant form is linked to mutations in the *HFE* gene and constitutes the most frequent genetic disorder in Caucasians. Other forms are caused by mutations in *transferrin receptor 2* (*TFR2*), hemojuvelin (*HJV*) or the hepcidin (*HAMP*) genes. Clinical complications of hemochromatosis include liver cirrhosis/HCC, cardiomyopathy, diabetes, endocrinopathy, arthritis and osteoporosis. Earlier studies have shown that hepatic iron overload promotes oxidative stress in humans [148,149] and mouse models [142,150], consistent with the role of iron in catalyzing ROS production and propagation via Fenton chemistry [24]. This is thought to drive liver fibrosis and hepatocarcinogenesis. Nevertheless, it is possible that other factors, such as inflammation, potentiate iron-induced liver injury and hepatotoxicity [151].

Wilson’s disease (hepatolenticular degeneration) is an autosomal recessive genetic defect of copper (Cu) metabolism that originates from mutations in the *ATP7B* gene [152]. The gene product, ATP7B, is a member of the cation-transporting P-type ATPase family. It is required for the biliary secretion of copper from hepatocytes as well as the transfer of copper from hepatocytes to plasma for its incorporation into apo-ceruloplasmin, a multicopper ferroxidase. Functional inactivation of ATP7B results in excessive deposition of copper in the liver and other tissues, which causes oxidative stress due to copper’s redox reactivity, and accounts for Wilson’s disease-associated hepatotoxicity [149,153]. Unshielded copper promotes oxidative stress via Fenton chemistry, pretty much like iron. Thus, copper overload in the liver of Wilson’s disease patients is toxic and, if untreated or late diagnosed, can lead to steatosis, inflammation, fibrosis, cirrhosis, and in some cases progression to fulminant hepatic failure or HCC [154].

Alpha-1 antitrypsin deficiency is a hereditary proteinopathy that can increase the risk of lung and/or liver disorders in children and adults [155]. Alpha-1 antitrypsin (A1AT), a member of the serine protease inhibitors (serpin) superfamily, is abundantly produced by hepatocytes and secreted into the bloodstream [156]. It constitutes the predominant circulating serpin in human plasma and serves to protect tissues against enzymatic destruction by proinflammatory proteases such as neutrophil elastase. Different mutations have been identified in the A1AT coding gene (*SERPINA1*), among which the Z variant is mainly associated with liver diseases [157]. Z-A1AT protein with abnormal folding aggregates in the ER of hepatocytes, instead of being efficiently secreted. This promotes ER stress, mitochondrial damage, necroinflammation and oxidative stress, which increase susceptibility to liver fibrosis, cirrhosis and HCC [158]. The role of oxidative stress in the development of liver disease has also been described in a murine model of A1AT deficiency [159].

## 10. Oxidative Stress in Liver Fibrosis

Hepatic fibrogenesis is defined as a natural wound-healing response to hepatocellular damage, characterized by the increased production and accumulation of ECM to encapsulate and isolate the injured regions of liver tissue for repair [160]. The dysregulation of the wound-healing process due to lack of inhibition or insufficient elimination of harmful agents can lead to sustained and uncontrolled tissue repair responses. These are associated with pathological changes in the ECM, formation of fibrous scar and ultimately liver fibrosis, a complex but potentially reversible process [161]. Measurement of the amount of fibrosis in liver sections is called staging. There are five stages: F0, no scarring (physiological tissue); F1, minimal scarring; F2, scarring extending outside the liver area (significant fibrosis); F3, fibrosis spreading and forming bridges with other fibrotic liver areas (severe fibrosis); and F4, cirrhosis or advanced scarring.

Liver fibrosis occurs in response to chronic liver injury and inflammation triggered by various factors such as alcohol abuse, NAFLD, hepatitis B or C, autoimmune hepatitis, infection with Schistosoma parasite, toxin- or drug-induced hepatotoxicity, genetic liver disorders or cholestatic disease. Unrecognized or untreated liver fibrosis can have severe consequences and eventually progress to end-stage liver cirrhosis and HCC. The initiation and progression of liver fibrosis occur as a result of interaction between different types of cells, including parenchymal and non-parenchymal liver-resident cells, as well as recruited immune cells to the liver [160].

Activated HSCs are the major producers of ECM proteins in the injured liver (Figure 4) [18]. HSCs activation occurs in response to a series of events including the release of ROS and inflammatory mediators from damaged hepatocytes, activation of inflammatory cells such as KCs to secrete profibrotic cytokines, lymphocyte infiltration into the injured site, and proliferation of cholangiocytes (ductular reaction). Under these conditions, LSECs undergo morphological modifications such as capillarization, which inhibits perfusion between blood and liver cells [13].

In addition to activated HSCs, further cell types contribute to the production of ECM proteins and the development of fibrosis. These include portal fibroblasts, bone marrow-derived myofibroblasts, fibrocytes originating from bone marrow hematopoietic cells, and myofibroblasts derived from liver epithelial cells through the epithelial–mesenchymal transition (EMT) process [162]. During fibrogenesis, the loose ECM composed of mainly non-fibrous collagen types IV and VI and laminin turns into a dense matrix enriched in fibrogenic collagens, especially type I, as well as non-collagenous glycoproteins such as fibronectin and proteoglycans [163]. Therefore, significant changes in the amount and composition of ECM components located in the space of Disse disrupt the physiological architecture and function of the fibrotic liver.

At the biochemical level, liver fibrogenesis involves the TGF-β/SMAD, Wnt/β-catenin and Hedgehog (Hh) signaling pathways [164]. TGF-β is the most effective pro-fibrogenic cytokine that activates HSCs in a SMAD2/3-dependent manner and directly induces transcription of the collagen type I α1 (*COL1A1*) and collagen type I α2 (*COL1A2*) genes [165]. In addition, TGF-β enhances HSC activation via non-SMAD pathways (MAPK, ERK, p38 and JNK). Connective tissue growth factor (CTGF), IL-6, TNFα or IL-1β can have a synergistic interaction with TGF-β in liver fibrosis [166,167,168]. However, IL-6 and IL-17 induce *COL1A1* transcription via the STAT3 signaling pathway [169,170]. Under pathological conditions, cytokines such as IL-33 can be released from the stressed hepatocytes to activate HSCs and promote fibrosis [171]. Gelatinase is also released by activated KCs and causes phenotypic changes in ECM by degrading collagen type IV [172]. Activated HSCs produce fibronectin, TGF-β and PDGF that lead to liver fibrosis. PDGF can increase the expression of tissue inhibitors of metalloproteinases (TIMPs) and inhibit collagenase activity, thereby increasing ECM deposition [173].

Macrophages modulate liver fibrosis by providing an inflammatory milieu that favors extensive production of pro-inflammatory cytokines and chemokines, which in turn activate HSCs [160,161]. M1 macrophages may convert to the anti-inflammatory M2 subtype, critical for the progression of liver fibrosis. Thus, M2 macrophages produce pro-fibrogenic factors such as TGF-β, vascular endothelial growth factor (VEGF) and galectin-3, which promote myofibroblast proliferation and activation leading to ECM deposition. Sustained fibrosis can cause the production of growth factors, proteolytic enzymes, pro-fibrogenic cytokines and collagen fragments [160,161].

Factors that trigger the development of chronic liver disorders such as ethanol abuse, accumulation of free fatty acids, or iron overload can enhance the production of ROS. There is experimental and clinical evidence that oxidative stress is involved in the initiation and progression of liver fibrosis, while crosstalk between pathways of oxidative stress and liver fibrogenesis is well established [174,175]. Thus, ROS may trigger the death of hepatocytes, intensify inflammatory responses, stimulate the release of pro-inflammatory cytokines from KCs and immune cells, and directly activate HSCs to produce pro-fibrogenic molecules. A major source of ROS in the context of liver fibrosis are the NOX enzymes, and especially the NOX1, NOX2 and NOX4 isoforms. Their importance was determined by experiments with NOX inhibitors such as diphenylene iodonium (DPI), as well as with mouse models bearing genetic disruption of the p47phox regulatory subunit of NOX enzymes [81,82,83,84,85]. Oxidative stress is particularly relevant in models of experimental liver fibrosis in response to bile duct ligation (BDL) or chronic exposure of animals to carbon tetrachloride (CCl_4_) or thioacetamide (TAA) [176].

Targeting oxidative stress has a therapeutic potential in liver disease, and novel drugs against mitochondrial dysfunction, ER stress or NOX activity have shown promising results in preclinical settings of NAFLD, NASH and liver fibrosis [38,175,177]. In general, antioxidant drugs and supplements are expected to fortify the antioxidant capacity of the liver and antagonize pathogenic oxidative stress responses. Plant antioxidant compounds, such as polyphenols and flavonoids, and nutritional antioxidants, such as zinc and coenzyme Q10, can induce antioxidant enzymes (CAT, SOD and GPX) [38,178]. Other drugs can affect the levels of non-enzymatic antioxidants. Thus, NAC increases GSH concentration in hepatocytes; this reduces ER stress, improves mitochondrial function, and protects against acute liver injury [179]. Nevertheless, the clinical efficacy of antioxidant drugs and supplements in the context of chronic liver diseases and liver fibrosis remains to be established [38,175,177,180]. Clinical data obtained thus far suggest that vitamin E offers the most effective antioxidant therapeutic tool that can ameliorate metabolic liver disease in some patients, but not prevent or reverse liver fibrosis (see Section 8).

Theoretically, liver fibrosis could be treated by targeting HSCs. The rationale is that during the recovery phase from liver injury, activated HSCs are cleared via cellular senescence and apoptosis, or revert to an inactive state [18,161]. Thus, drugs with the capacity to induce the clearance of activated HSCs could be used to promote regression of liver fibrosis in chronic liver disease.

While liver fibrosis is considered reversible, at least if treated in early stages, it may progress to cirrhosis (stage F4), representing advanced states of the disease [181]. Liver cirrhosis is characterized by the replacement of the normal liver parenchyma with regenerative hepatic nodules surrounded by fibrotic scar tissue, which eventually leads to the loss of normal liver function [182]. This can cause complications such as portal hypertension (increased pressure in the portal vein) resulting in splenomegaly, hypersplenism and varices (enlarged blood vessels) in the esophagus and stomach. Varices are prone to bleeding and can be life-threatening. The development of liver cirrhosis is dynamic and encompasses an initial asymptomatic stage referred to as compensated cirrhosis, followed by a symptomatic phase known as decompensated cirrhosis with clinical manifestations ranging from the development of portal hypertension complications to liver failure [183]. Liver cirrhosis, regardless of its etiology, is recognized as the most potent risk factor for HCC; thus, 80–90% of patients with HCC have underlying cirrhosis [184].

## 11. Oxidative Stress in Hepatocellular Carcinoma

Liver cancer is the sixth most frequently diagnosed type of cancer and accounts for the third leading cause of cancer mortality worldwide [185,186]. HCC comprises 75–85% of liver cancers and arises from malignant transformation of hepatocytes. Other histological subtypes of liver cancer include cholangiocarcinoma (CCA), which arises from the malignant transformation of cholangiocytes, and the rare hepatic angiosarcoma and hepatoblastoma. HCC is prevalent in both developing and developed countries and has the highest incidence in Africa and Asia [187]. It is characterized by sex disparity with 2–3 times prevalence in males [188] and has relatively poor prognosis compared to other solid tumors with a 5-year survival of 18% in advanced HCC [189].

ROS can cause genomic instability and mutations either via direct DNA oxidation, or indirectly by inducing DNA damage [190]. Oncogenic activation and/or inactivation of tumor suppressor genes are subsequent steps that initiate carcinogenesis. It is well established that a moderate increase in cellular ROS levels is important in cellular transformation by activating the signaling cascades related to cancer cell survival, including the MAPK/ERK1/2 pathways [191].

Oxidative stress promotes HCC via genetic, but also epigenetic alterations, such as changes in the expression of oncogenes, tumor suppressors and proinflammatory genes [190]. Common HCC-related mutations have been identified in genes involved in p53, Wnt and Retinoblastoma-1 (RB1) pathways [192]. A major lipid peroxidation product, namely, 4-hydroxy-trans-2-nonenal (HNE), forms DNA adducts and can lead to p53 mutations that are associated with HCC [193]. High ROS levels facilitate epithelial-to-mesenchymal transition (EMT) in HCC via epigenetic hypermethylation of the E-cadherin gene promoter, leading to its suppression [194].

Oxidative stress also plays a key role in the pathogenesis and progression of HCC by modulating the expression of cytokines and growth factors involved in cancer cell survival [190]. Viral infections, excessive alcohol consumption, and lipids toxicity are important risk factors that may contribute to hepatocarcinogenesis via oxidative stress-related mechanisms [195,196]. Chronic inflammation, which predisposes hepatocytes to HCC [197], is often observed in liver diseases such as NASH, hepatitis B and hepatitis C.

Experiments in mouse models have established molecular links between oxidative stress, antioxidant defense systems and HCC. For instance, mice with targeted disruption of superoxide dismutase 1 (SOD1) [198] or the transcription factor NRF1 [199] (an NRF2 homolog), showed increased susceptibility to hepatocarcinogenesis. Cohort studies showed that low SOD2 expression is associated with reduced survival in HCC patients [200]. In another study, HCC patients exhibited decreased expression of the antioxidant enzyme glutathione-S-transferase P1 (GSTP1) in peripheral blood mononuclear cells (PBMCs) compared to chronic hepatitis B patients, which was associated with increased markers of oxidative stress [201].

The tight connection between oxidative stress and HCC is highlighted in the predisposition of hemochromatosis patients to hepatocarcinogenesis [202,203,204,205], which is recapitulated in rodent models of iron overload [43,206,207]. There is evidence that hepatic iron overload promotes oxidative stress via the Fenton reaction and by accelerating lipid peroxidation, which leads to DNA damage and mutations in the tumor suppressor p53 [149,205].

The causative role of NOX enzymes in HCC mediated by fibrosis and inflammation is well established [208]. During hepatocarcinogenesis in mice, NOX enzymes are major sources of ROS production in liver cells [209]. Their activity can disrupt redox signaling pathways involved in the initiation and progression of HCC. Different NOX isoforms show differential effects in HCC development [210,211]. Thus, NOX4 may have a protective role against HCC by promoting TGF-β-induced senescence in tumor cells [212]. However, NOX1 stimulates the growth of HCC cell lines via epidermal growth factor receptor (EGFR) signaling [213]. NOX1 expression in liver macrophages can promote liver tumorigenesis by increasing the production of inflammatory cytokines [214]. In line with the experimental data, NOX1 overexpression correlates with poor prognosis in HCC patients, while opposite effects have been observed with NOX4 [210]. NOX1 is also critical for the development of experimental HCC following the treatment of rodents with diethylnitrosamine (DEN); thus, the pharmacological inhibition of NOX1 attenuated hepatocarcinogenesis by mitigating inflammatory, angiogenic and fibrogenic responses [209].

ER stress develops following the accumulation of unfolded or misfolded proteins in the ER and is a recognized contributor to hepatocarcinogenesis via the unfolded protein response (UPR) [215,216]. For instance, experiments in mice showed that ER stress causes HCC by activating NF-κB and TNF-α inflammatory signaling pathways [217]. ER stress and UPR are tightly linked to oxidative stress. ER stress can increase the production of ROS, especially via NOX enzymes [218,219]. On the other hand, since the protein folding process relies on redox homeostasis, oxidative stress can exacerbate ER stress by increasing misfolded protein production [220].

The role of the NRF2/ARE pathway in HCC has been extensively investigated. Even though adaptive NRF2 induction appears protective against NASH and other liver diseases [37], there is evidence that sustained NRF2 activation is maladaptive and promotes increased cancer cell proliferation, migration, metastasis and survival, as well as drug resistance in many cancers, including HCC [221,222,223]. The underlying mechanisms include induction of the anti-apoptotic factor Bcl-xL, induction of matrix metalloproteinase 9 (MMP-9) and inhibition of autophagy. In line with these data, high NRF2 expression correlates with poor prognosis in HCC patients [223]. Therefore, modulating the NRF2 pathway may have translational potential in HCC.

## 12. Conclusions

We discussed the effects of oxidative stress on various liver cell types and liver pathophysiology, as well as on liver disease pathogenesis and progression. The underlying mechanisms are complex and in many instances are tightly linked to inflammatory pathways. While the role of oxidative stress as a contributor to liver diseases is undisputed, it is not yet clear whether targeting oxidative stress pathways offers therapeutic benefits. This is certainly the case in acute liver injury due to acetaminophen toxicity, where the antioxidant N-acetylcysteine (NAC) is used as the first line of treatment. There is also evidence that vitamin E can improve liver function in some NAFLD/NASH patients. However, more work is needed to evaluate the potential of antioxidant therapies in the context of other liver diseases.

## Figures and Tables

**Figure 1 antioxidants-12-01653-f001:**
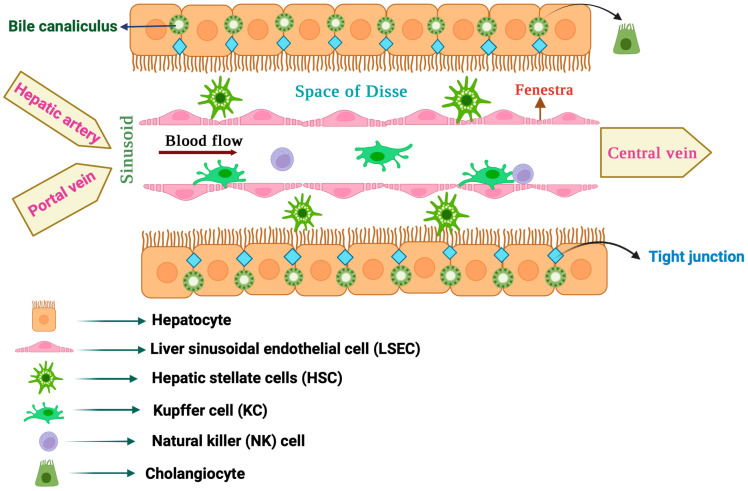
Anatomy of the liver lobule.

**Figure 2 antioxidants-12-01653-f002:**
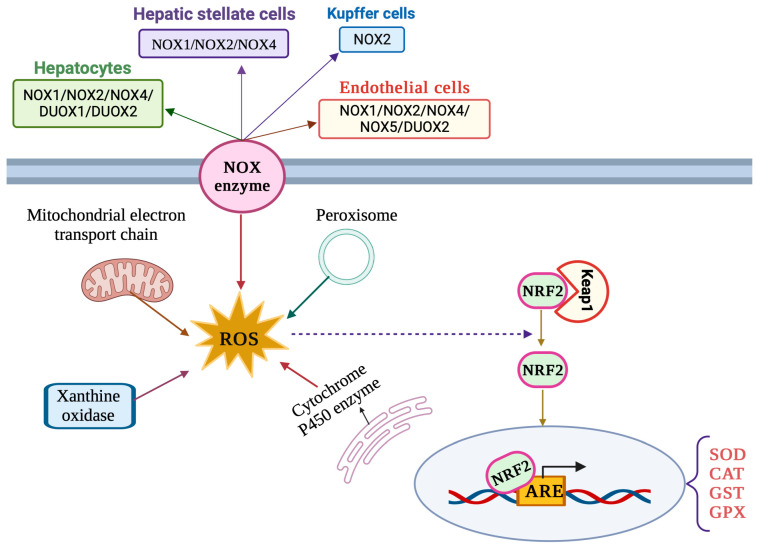
Major sources of ROS production and NRF2 signaling pathway in liver cells.

**Figure 3 antioxidants-12-01653-f003:**
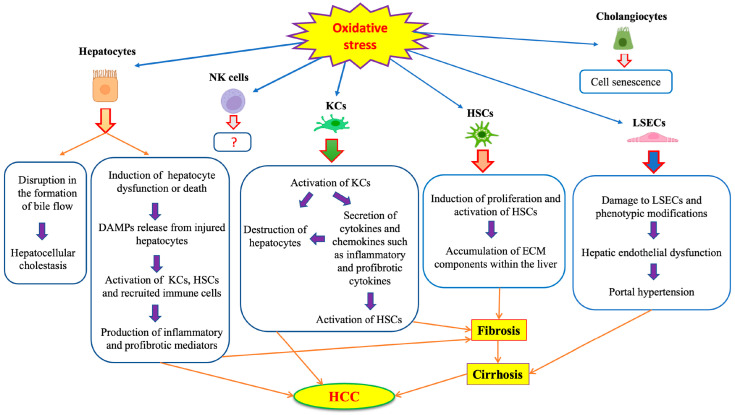
Effects of oxidative stress on different liver cells.

**Figure 4 antioxidants-12-01653-f004:**
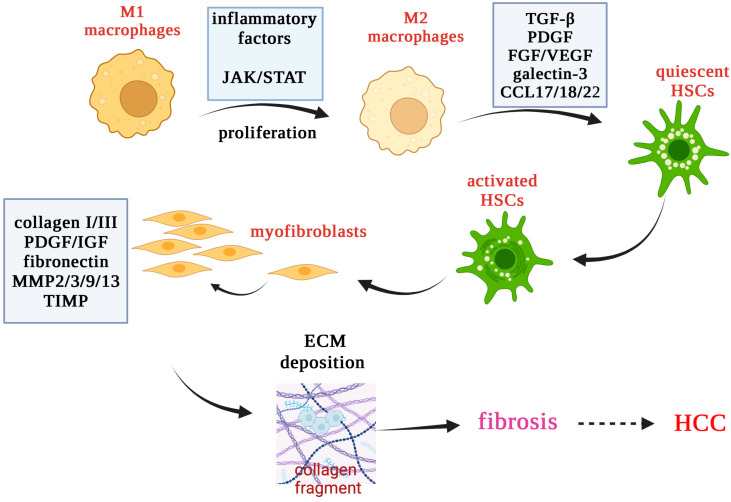
Role of inflammatory and pro-fibrotic cytokines in liver fibrosis that may further lead to HCC.

## Data Availability

Data sharing not applicable.

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
