# Peer review of "Oxidative Stress in Liver Pathophysiology and Disease"

_antioxidants, 2023, doi:10.3390/antiox12091653_

Round 1

Reviewer 1 Report

"Oxidative stress in liver pathophysiology and disease" article by Abdolamir Allameh et all., is an excellent manuscript, very well done and important in the field of hepatology and also in NAFLD/MAFLD.The scientific content sounds very good, and the graphical illustrations are very clear and comprehensive overall, the main conclusions are very important for clinicians.

Author Response

We thank reviewer 1 for the positive comments.

Reviewer 2 Report

This review paper addresses the role of oxidative stress in liver pathophysiology and disease. However, I have a major concern regarding the limited emphasis placed on the significance of oxidative stress in the progression of liver damage and the potential treatments involving antioxidants. Specifically, I would like to highlight the content in Section 10, "Oxidative stress in liver fibrosis," where most of the paragraphs and Figure 4 predominantly focus on immune system dysregulation. It is crucial for the authors to reinforce the importance of oxidative stress in the pathology progression of liver damage and the potential treatment options utilizing antioxidants.

Furthermore, there are some sentences within the paper that are difficult to comprehend. For instance, the sentence "While liver fibrosis is considered reversible, at least if treated in early stages, it may also progress to cirrhosis (stage F4), an irreversible form of the disease" and this sentence is not adequately supported by relevant references. Please check for the rest in this manuscript.

Author Response

This review paper addresses the role of oxidative stress in liver pathophysiology and disease. However, I have a major concern regarding the limited emphasis placed on the significance of oxidative stress in the progression of liver damage and the potential treatments involving antioxidants. Specifically, I would like to highlight the content in Section 10, "Oxidative stress in liver fibrosis," where most of the paragraphs and Figure 4 predominantly focus on immune system dysregulation. It is crucial for the authors to reinforce the importance of oxidative stress in the pathology progression of liver damage and the potential treatment options utilizing antioxidants.

We thank reviewer 2 for the feedback. A new paragraph has been added to section 10 (lines 542-553), addressing these issues.

Furthermore, there are some sentences within the paper that are difficult to comprehend. For instance, the sentence "While liver fibrosis is considered reversible, at least if treated in early stages, it may also progress to cirrhosis (stage F4), an irreversible form of the disease" and this sentence is not adequately supported by relevant references. Please check for the rest in this manuscript.

The sentence has been rephrased, as requested (lines 559-560).

Reviewer 3 Report

The manuscript “Oxidative stress in liver pathophysiology and disease” is a review article regarding the effects of oxidative stress on liver pathophysiology and the mechanisms by which oxidative stress may promote liver disease.

I really appreciate the work performed by authors. The manuscript is well written, provides a very good view on the topic and may be of interest for the readers. However, there are important concerns that authors must address:

1.     The manuscript requires an accurate language revision since are present some typos.

1.     The review seems to cover the actual literature only partially. Indeed, several important contributions to the topic have been ignored by authors. A paragraph regarding the importance of the enzyme nicotinamide n-methyltransferase (NNMT) in liver metabolism and alcoholic liver disease is mandatory, due to its high expression in liver, crucial role in NAD+ homeostasis (PMID: 36829935), involvement in alcoholic liver disease (PMID: 33340581; PMID: 32389809), since it may affect autophagy (PMID: 30093610; PMID: 32489327) and due to its putative protective role in endothelium which may counteract oxidative stress in liver vessels (PMID: 34153425).

2.     In general figures present a low quality and a reduced sharpness which makes difficult to read them. Their quality should be improved.

3.     Figure 3: it contains typos. For example “senescenece”, “desruction". Moreover, the use of mixed dotted arrows and arrows makes the figure looking worse.

4.     Figure 4: TGF-B should be replaced by TGF-β. Moreover, it should be consistent with the written text, where authors refers to it with “TGFβ”. Please reconcile.

5.     The paragraph 4 should be improved. Indeed, authors should report the effect of oxidative stress on oxidation of spectrin in liver sinusoidal endothelial cells, which alters the functionality of fenestrae, thus contributing to liver disease  (PMID: 31569283).

Few typos detected.

Author Response

The manuscript “Oxidative stress in liver pathophysiology and disease” is a review article regarding the effects of oxidative stress on liver pathophysiology and the mechanisms by which oxidative stress may promote liver disease.

I really appreciate the work performed by authors. The manuscript is well written, provides a very good view on the topic and may be of interest for the readers.

We thank reviewer 2 for the positive comments and the feedback.

However, there are important concerns that authors must address:

  1. The manuscript requires an accurate language revision since are present some typos.

We revised the manuscript to correct typos, as requested.

  1. The review seems to cover the actual literature only partially. Indeed, several important contributions to the topic have been ignored by authors. A paragraph regarding the importance of the enzyme nicotinamide n-methyltransferase (NNMT) in liver metabolism and alcoholic liver disease is mandatory, due to its high expression in liver, crucial role in NAD+ homeostasis (PMID: 36829935), involvement in alcoholic liver disease (PMID: 33340581; PMID: 32389809), since it may affect autophagy (PMID: 30093610; PMID: 32489327) and due to its putative protective role in endothelium which may counteract oxidative stress in liver vessels (PMID: 34153425).

A new paragraph addressing this important issue has been added in lines 363-374.

  1. In general figures present a low quality and a reduced sharpness which makes difficult to read them. Their quality should be improved.

All figures were replaced with high resolution images.

  1. Figure 3: it contains typos. For example “senescenece”, “desruction". Moreover, the use of mixed dotted arrows and arrows makes the figure looking worse.

The figure was updated to address these issues.

  1. Figure 4: TGF-B should be replaced by TGF-β. Moreover, it should be consistent with the written text, where authors refers to it with “TGFβ”. Please reconcile.

We now use TGF-β throughout the text and figures.

  1. The paragraph 4 should be improved. Indeed, authors should report the effect of oxidative stress on oxidation of spectrin in liver sinusoidal endothelial cells, which alters the functionality of fenestrae, thus contributing to liver disease (PMID: 31569283).

Done (lines 244-247).

Round 2

Reviewer 3 Report

The authors addressed all the raised issues and therefore the manuscript can be published.

English is generally fine.